# 3D GPR Image-based UcNet for Enhancing Underground Cavity Detectability

**Man-Sung Kang [1], Namgyu Kim [2], Seok Been Im [3], Jong-Jae Lee [2] and Yun-Kyu An [1],\***

1   Department of Architectural Engineering, Sejong University, Seoul 05006, Korea; kms102353@sju.ac.kr
2   Department of Civil and Environmental Engineering, Sejong University, Seoul 05006, Korea; namgyu.kim@sejong.ac.kr (N.K.); jongjae@sejong.ac.kr (J.-J.L.)
3   Research Institute for Infrastructure Performance, KISTEC, Jinju 52856, Korea; sbeeni@kistec.or.kr
\*   Correspondence: yunkyuan@sejong.ac.kr; Tel.: +82-2-6935-2426

**Abstract:** This paper proposes a 3D ground penetrating radar (GPR) image-based underground cavity detection network (UcNet) for preventing sinkholes in complex urban roads. UcNet is developed based on convolutional neural network (CNN) incorporated with phase analysis of super-resolution (SR) GPR images. CNNs have been popularly used for automated GPR data classification, because expert-dependent data interpretation of massive GPR data obtained from urban roads is typically cumbersome and time consuming. However, the conventional CNNs often provide misclassification results due to similar GPR features automatically extracted from arbitrary underground objects such as cavities, manholes, gravels, subsoil backgrounds and so on. In particular, non-cavity features are often misclassified as real cavities, which degrades the CNNs' performance and reliability. UcNet improves underground cavity detectability by generating SR GPR images of the cavities extracted from CNN and analyzing their phase information. The proposed UcNet is experimentally validated using in-situ GPR data collected from complex urban roads in Seoul, South Korea. The validation test results reveal that the underground cavity misclassification is remarkably decreased compared to the conventional CNN ones.

**Keywords:** ground penetrating radar; underground cavity detection network; deep convolutional neural network; automated underground object classification; phase analysis; super-resolution

## 1. Introduction

A series of sudden sinkhole collapses continuously occur on complex urban areas over the world. Recently, several sinkhole activities have been reported globally, such as the United States [1–3], China [4,5], Japan [6], and South Korea [7]. These sinkholes have resulted in a major disruption of traffic flow and utility services causing significant economic losses as well as critical human injuries and fatalities [8,9]. Since these sinkholes have shown up without any forewarning, there is an increasing demand for their early detection especially in complex urban areas.

Recently, ground penetrating radar (GPR) has been widely employed for early detection of underground cavities, which are most likely propagating to sinkholes, thanks to its fast scanning speed, nondestructive inspection, and 3D imaging capabilities [10–12]. GPR transmitters emit electromagnetic waves into the underground at several spatial positions along the scanning direction, and GPR receivers measure the reflected signals to establish 2D GPR image called a radargram, also known as a B-scan image. If the multi-channel GPR transmitters and receivers parallel to the scanning direction are equipped, 3D GPR images including B- and C-scan images can be obtained at once. Since the electromagnetic waves propagating along the underground medium are dominantly reflected from the abrupt change of electromagnetic permittivity, the reflection signal features appear in

the B- and C-scan images. To enhance visibility and detectability of the reflection signal features, a number of data processing techniques such as time-varying gain [13], subtraction [14,15], and basis pursuit-based background filtering [16] have been proposed. However, these techniques are highly susceptible to measurement noises especially in complex urban roads and sometimes unreliable due to decision making based on experts' experiences. Moreover, expert-dependent data interpretation becomes time consuming and cumbersome, as the amount of 3D GPR data increases.

To overcome the technical limitations, several researchers have made efforts to automate the GPR data classification process. Simi et al. proposed a Hough transform-based automatic hyperbola detection algorithm to reduce GPR data analysis time [17]. Li et al. utilized a randomized Hough transform to effectively find the parabola parameters [18]. In addition, histograms of oriented gradient features-based GPR data classification were proposed for automatically detecting underground objects [19,20]. More recently, neural networks and convolutional neural networks (CNNs) were widely used as a promising tool for automated GPR image recognition and classification. Mazurkiewicz et al. tried to identify underground objects using neural networks while reducing the processing time and human intervention [21–23]. Then, Al-Nuaimy et al. utilized both neural network and pattern recognition methods to automatically detect the buried objects [24]. Lameri et al. also applied CNN to detect landmines with pipeline B-scan images [25].

Although the previous studies have focused on the use of GPR B-scan images, it is often difficult to classify a specific target underground object by using only GPR B-scan images. In particular, the GPR B-scan images often tend to be similar among various underground objects such as cavities, manholes, pipes, electrical lines, gravels, concrete blocks and so on in complex urban areas. To enhance the classification performance, additional GPR C-scan images were simultaneously considered during the classification procedure using CNN [26,27]. Although the developed CNN using the combination of B- and C-scan images increased the classification accuracy compared to the conventional CNN using only B-scan images, it turned out that it still has difficulty differentiating underground cavities from chunks of gravel especially in complex urban areas due to their similar GPR reflection features. To address the misclassification issue caused by the underground chunks of gravel, Park et al. recently proposed a phase analysis technique of GPR data [16]. However, the temporal and spatial resolutions of the 3D GPR data are often insufficient for the precise phase analysis. Since the phase analysis results highly depend on a few pixel differences of the GPR images, the lack of GPR image resolution may cause false alarms during the phase analysis.

In this paper, a 3D GPR image-based underground cavity detection network (UcNet), which consists of CNN and the phase analysis of super-resolution (SR) GPR images, is newly proposed to enhance underground cavity detectability. By retaining the advantages of both CNN and phase analysis, underground cavities can be automatically classified with minimized false alarms. In particular, a deep learning-based SR network used for GPR image resolution enhancement significantly reduces the misclassification between the underground cavity and chunk of gravel. To examine the performance of the proposed UcNet, comparative study results on cavity detectability between the conventional CNN and the newly proposed UcNet are presented using in-situ 3D GPR data obtained from complex urban roads in Seoul, South Korea.

This paper is organized as follows. Section 2 explains the proposed UcNet, which is comprised of CNN, SR image generation, and phase analysis. Then, the 3D GPR data collection procedure from urban roads and experimental validation are described in Section 3. In particular, the comparative study results between the conventional CNN and newly proposed UcNet are addressed. Finally, Section 4 concludes the paper with a brief discussion.

## 2. Development of UcNet

Once 3D GPR data are obtained from a target area with various underground objects, the corresponding 2D GPR grid images comprised of the B- and C-scan images can be reconstructed for network training and testing as displayed in Figure 1 [26]. Figure 1a shows the representative 2D grid image of the cavity case, which have parabola and circle features that can be observed on

the B- and C-scan images, respectively. The similar features are also revealed in the manhole case of Figure 1b, although they have a different pattern and amplitude. On the other hand, no significant features appear in the subsoil background case as shown in Figure 1c, because there is no abrupt permittivity change. Thanks to these distinguishable features, underground objects can be classified well with the conventional CNNs. However, the underground gravel case of Figure 1d shows the similar morphological GPR B- and C-scan features as the cavity ones, which may cause false alarms. Thus, UcNet is newly proposed to minimize the false alarms.

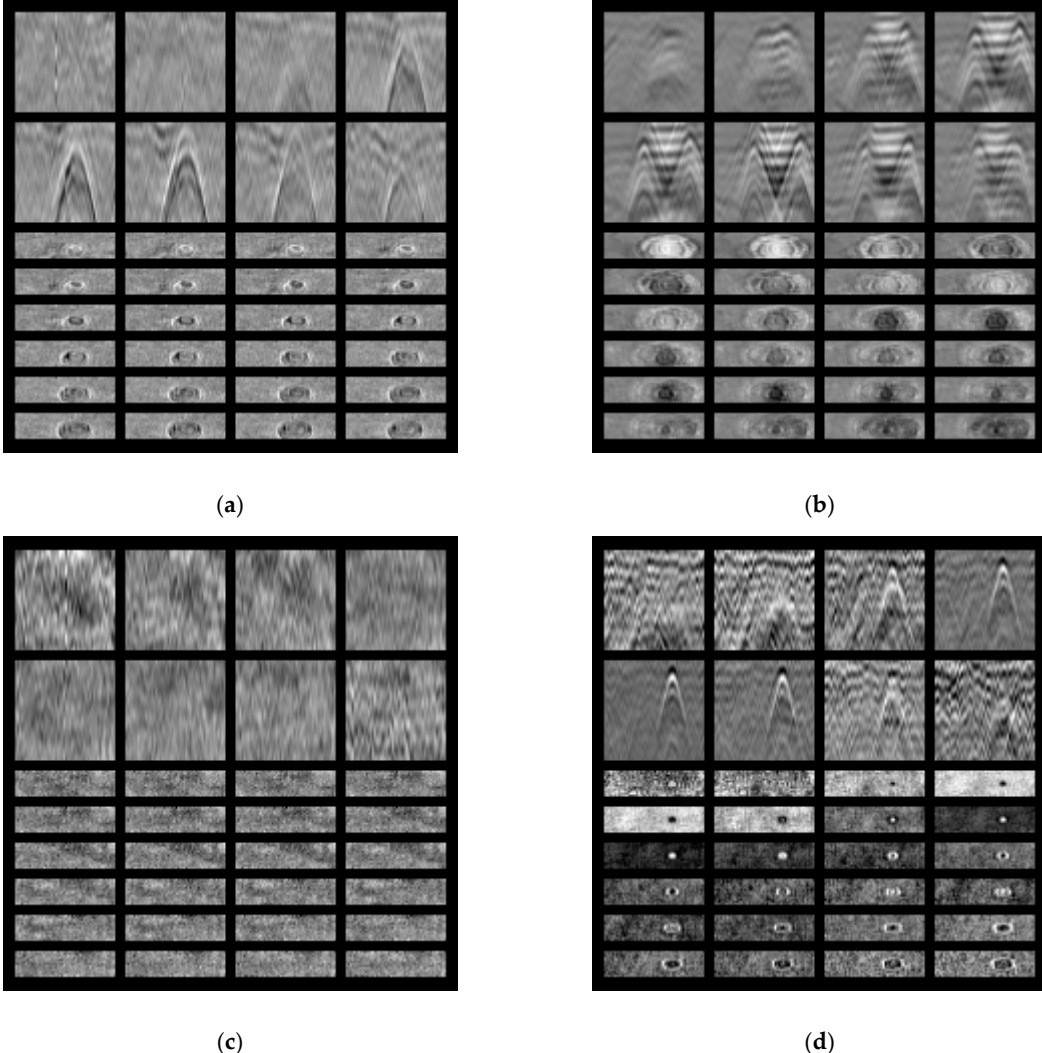

(**a**)          (**b**)

(**c**)          (**d**)

**Figure 1.** Representative 2D ground penetrating radar (GPR) grid images reconstructed from 3D GPR data: (**a**) cavity, (**b**) manhole, (**c**) subsoil background, and (**d**) gravel.

Figure 2 shows the overview of proposed UcNet consisted of the three phases, i.e., Phase I: CNN, Phase II: SR image generation, and Phase III: Phase analysis. First, the reconstructed 2D GPR grid images are fed into CNN for data classification. Subsequently, the cavity and gravel data classified in Phase I are transmitted into Phase II to generate their SR images from original low-resolution (LR) images. Finally, the phase analysis of the SR images is conducted in Phase III to update the classification results obtained in Phase I. Through these sequential processes, misclassification between the cavity and gravel cases can be minimized. The details of each phase are as follows.

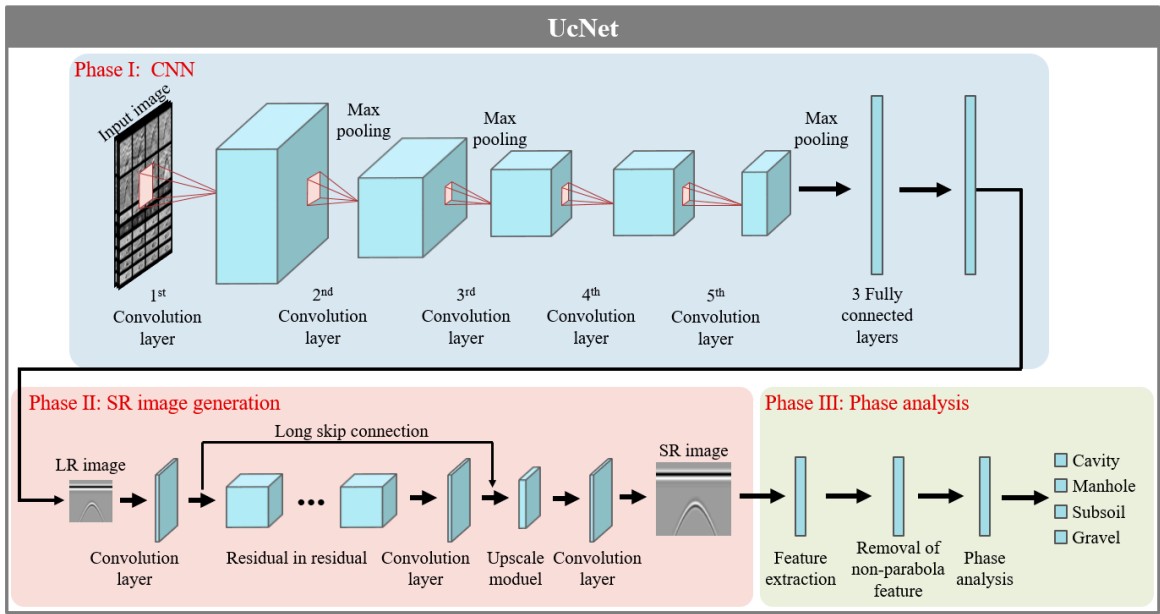

**Figure 2.** Overview of underground cavity detection network (UcNet): LR and SR denote the low-resolution and super-resolution, respectively.

Phase I: CNN is established by transfer learning from AlexNet [28], which is one of the widely used pre-trained CNN models for image classification, in this study. The modified CNN consists of five convolutional layers, three fully connected layers, three max pooling layers and 1000 softmax neurons, containing 650,000 neurons and 60 million parameters. To train CNN, the 2D GPR grid images are fed to the input layer consisted of the image size of $227 \times 227 \times 3$ pixels, and the pixel features are then extracted through the convolutional layers. Figure 3 shows the representative training GPR 2D grid images. The 1st convolutional layer uses the kernel of $11 \times 11 \times 3$ pixels with a stride of four. Consequently, the layer creates 96 feature maps and has the output of $55 \times 55 \times 96$ pixels. The max pooling layer, which is one of sub sampling techniques, is then arranged after the 1st convolutional layer to reduce the size of feature maps. Next, the 2nd convolution layer is operated with the kernel of $5 \times 5$ pixels and creates $27 \times 27 \times 256$ pixels. The max pooling layer is again arranged after the 2nd convolutional layer. The following convolutional layers are operated with the kernel of $3 \times 3$ pixels and create $13 \times 13 \times 256$ pixels. The max pooling layer is once again arranged after the 5th convolutional layer. Once the features are extracted and shrunken on the convolutional layers, the feature maps are fed to fully connected layers. To avoid an overfitting issue, dropout layers are arranged after 1st and 2nd fully connected layers. In addition, the rectified linear unit is selected as an activation function. Finally, the output of the last fully connected layer is fed into a softmax layer having four probabilities, i.e., cavity, manhole, subsoil background, and gravel in this study.

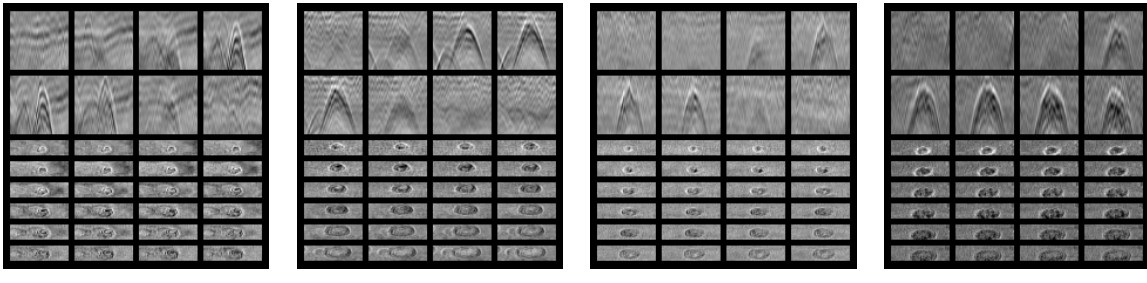

(**a**)

**Figure 3.** *Cont.*

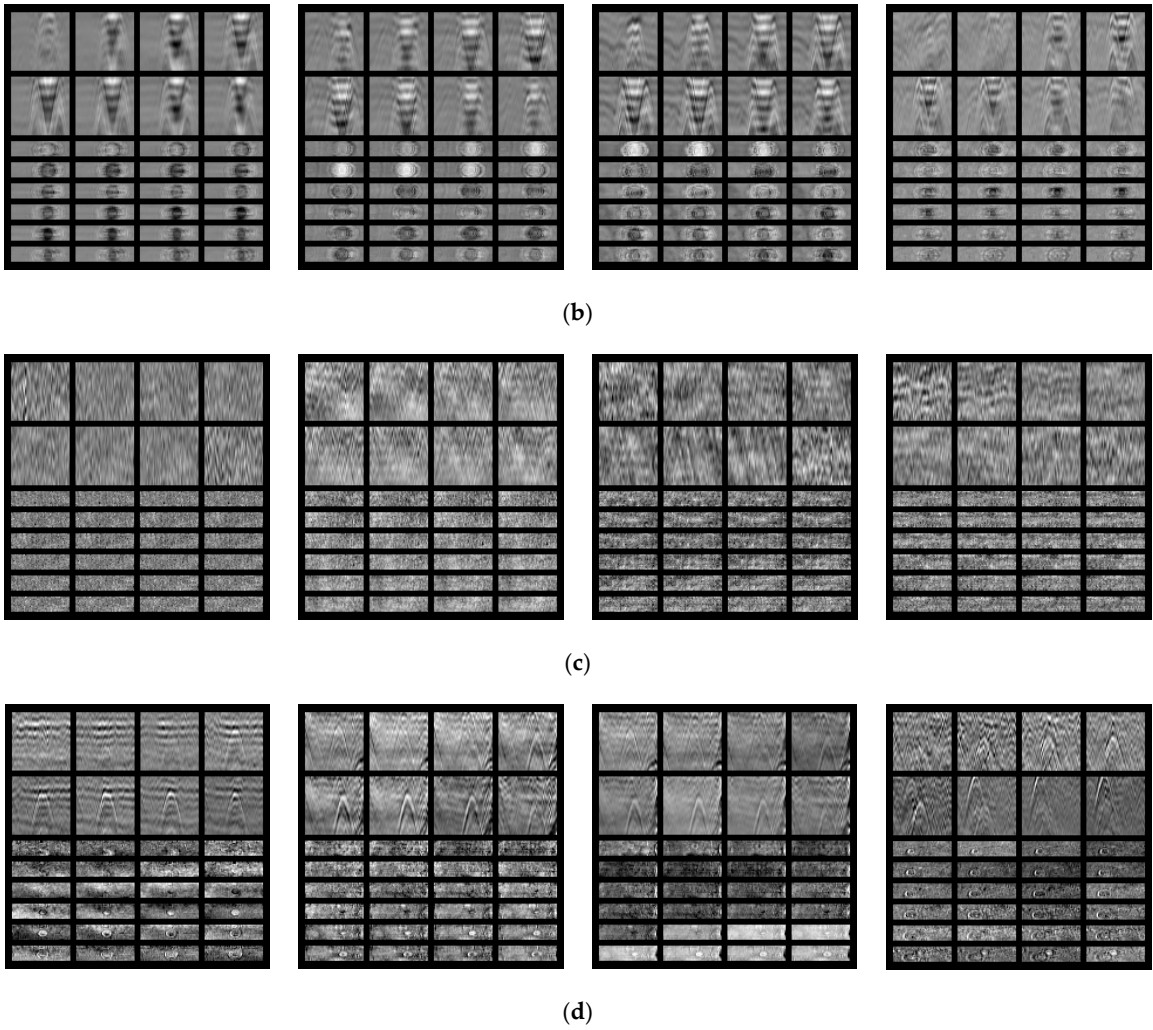

**Figure 3.** Representative training 2D GPR grid images of a (**a**) cavity, (**b**) manhole, (**c**) subsoil background, and (**d**) gravel.

Phase II: Once the underground objects are classified by Phase I, the cavity and gravel images are automatically transferred to Phase II. The reconstructed GPR grid image is comprised of eight B-scan and 24 C-scan images. The original sizes of B- and C-scan images are $50 \times 50$ and $50 \times 13$, respectively. To proceed Phase II, the 5th representative B-scan image is selected from each reconstructed GPR 2D grid image. Since the resolution of the selected B-scan image is not enough for Phase II as shown in Figure 4, the subsequent SR image generation process is necessary.

The SR image generation network was constructed by using the residual channel attention network [29], which is one of the deep learning networks comprised of 500 layers and 1.6 M parameters for image resolution enhancement. This network utilizes the residual in residual structure and the channel attention mechanism to enhance the feature learning of high frequency channels, which is useful for reconstructing high-resolution images among the various channels that make up image data. The residual channel attention network consists of the four main parts, i.e., the convolution layer, residual in residual structure, upscale module, and last convolution layer. First, the convolution layer is shallow feature extraction for the input image. Then, the residual in residual structure extracts deep features through the high frequency information learning. The residual in residual structure is the very deep structure comprised of 10 residual groups, and each residual group consists of 20 residual blocks. Each residual group is connected by a long skip connection as shown in Phase II of Figure 2. This residual in residual structure allows the residual channel attention network to learn more effective high frequency information by skipping the low frequency through the skip connection. The shallow

and deep feature data, which have passed through each residual block and group, are extended to the SR size through the upscale module. The upscale module is composed of a deconvolution layer consisting of 256 kernels with 3 × 3 size and a single stride, which increases the size of each pixel by four times in this study. Finally, it is restored as the SR image through the last convolution layer. The 800 training images from the DIV2K dataset, which is well known high-quality images with 2K resolution, were used as the training dataset [30]. Then, 100 images collected from Urban100 dataset were used as the validation dataset. The representative resultant GPR images between LR (50 × 50 pixels) and SR (200 × 200 pixels) are compared in Figure 5. The SR image is successfully generated without any information loss.

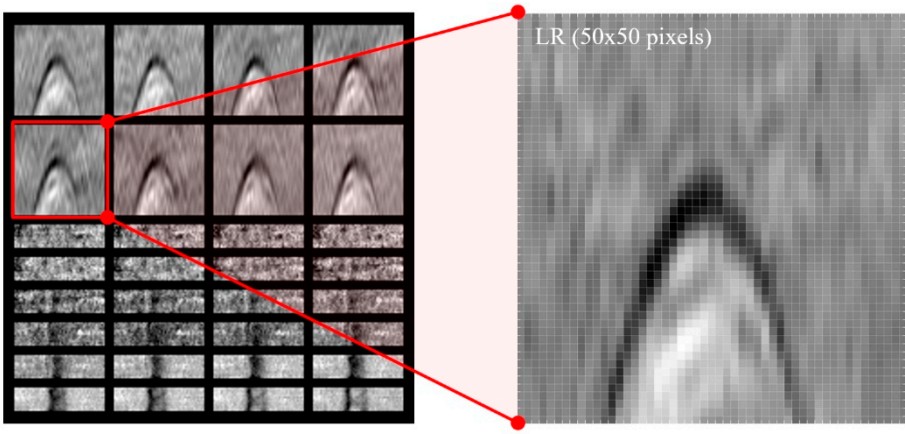

**Figure 4.** Representative LR GPR image for SR image generation input.

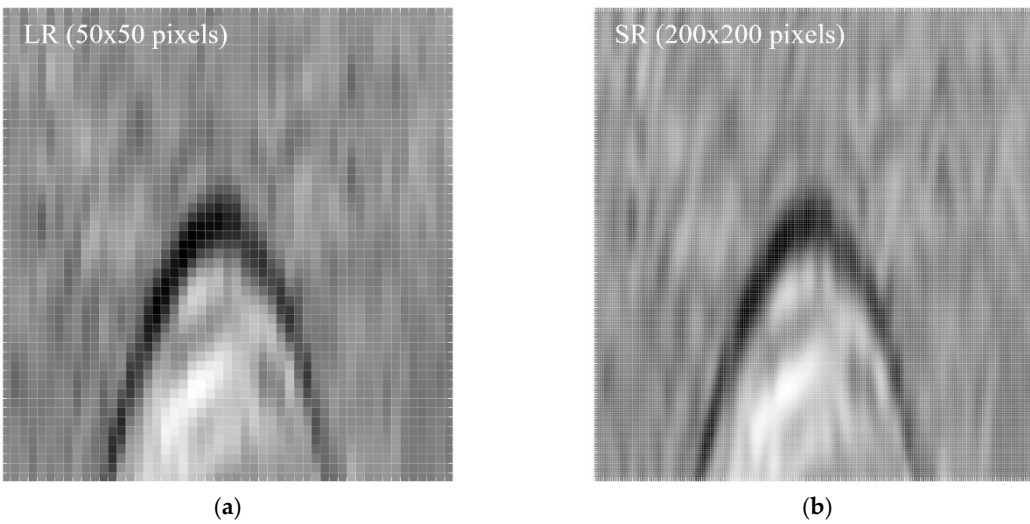

**Figure 5.** GPR image enhancement results: (**a**) LR and (**b**) SR images.

Phase III: Once the SR GPR images are generated in Phase II, the phase analysis is subsequently carried out in Phase III. To automate the phase analysis of the SR images, the feature extraction, removal of non-parabola features, and parabola boundary extraction are continuously proceeded as shown in Phase III of Figure 2. Figure 6a shows the representative SR B-scan image obtained from Phase II. Since the B-scan image includes a number of noise components, the feature extraction with noise removal procedure is sequentially performed. First, a median filter is applied to the B-scan image for removing pepper noise components, and the filtered image is then normalized with respect to the maximum amplitude. Subsequently, the extreme value distribution with 95% confidence interval is applied to obtain the dominant features. However, since the remaining features still contain non-parabola boundaries, as shown in Figure 6b, it should be removed. The unwanted spotted

and non-continuous features can be removed by eliminating non-continuous pixels less than 500, as displayed in Figure 6c. Note that the target detectable cavity size of 30 cm is equivalent to 500 pixels in the SR image. Next, the parabola feature can be extracted by using the gradient between the remaining objects' centroid and the extremum of left- and right-side pixels. Since the empirical gradient value of typical parabolas is larger than 20°, only the parabola feature remains, as displayed in Figure 6d.

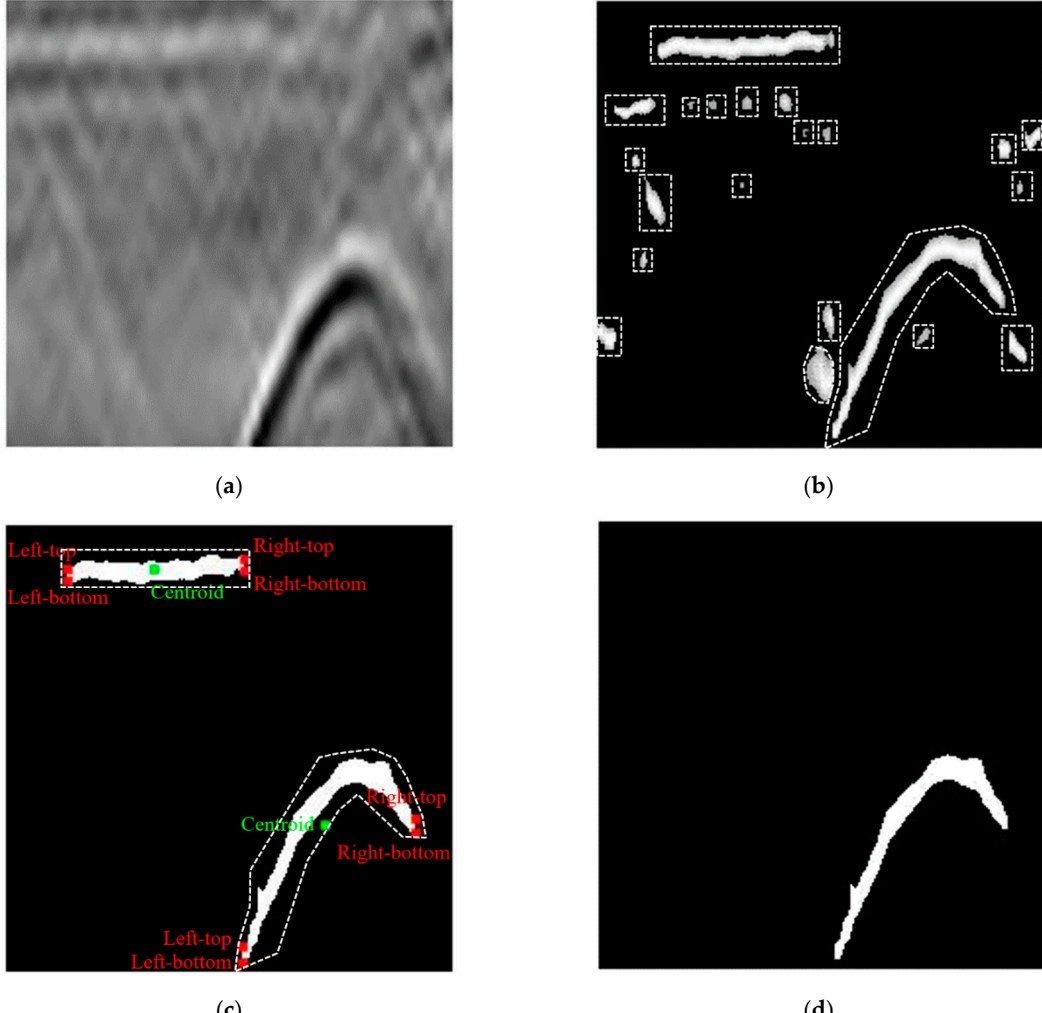

**Figure 6.** Representative (**a**) SR B-scan image, (**b**) feature extracted image, (**c**) noise removal image, and (**d**) parabola boundary extraction image.

Once the parabola boundary is automatically extracted, its phase can be analyzed as the final procedure. The extracted boundary value is converted to phase information using Equations (1) and (2).

$$H(x,z) = \frac{1}{\pi} P \int_{-\infty}^{\infty} \frac{I(x,z)}{z-\tau} d\tau, \tag{1}$$

where $P$ denotes the Cauchy principal value. $I(x,z)$ is the GPR A-scan data. $x$ and $z$ are the spatial coordinates along the scanning and depth directions, respectively. The instantaneous phase value at each spatial point can be calculated by:

$$\theta(x,z) = \tan^{-1}\left(\frac{Im[H(x,z)]}{Re[H(x,z)]}\right), \tag{2}$$

where *Re* and *Im* represent the real and imaginary components of a complex value, respectively.

If the relative permittivity of the underground object is lower than that of the surroundings, the reflected electromagnetic waves are in-phase with the radiated waves. Otherwise, the reflected electromagnetic waves will be indicated out-of-phase [16]. The phase change ratio of the extracted parabola boundary can be expressed by:

$$\Delta\theta = \frac{\theta(x, lp) - \theta(x, fp)}{lp - fp},$$ (3)

where $fp$ and $lp$ denote the first and last pixels along the A-scan of the extracted phase boundary.

To numerically validate the phase analysis of Phase III, the cavity and gravel cases were modeled using gprMax [31] as shown in Figure 7a. The target model was comprised of $1 \times 0.8$ m$^2$ soil layer and $1 \times 0.2$ m$^2$ air layer. Then, the underground cavity and gravel were respectively inserted inside the soil layer with a depth of 0.5 m as depicted in Figure 7a. Here, the relative permittivity values of the soil, cavity and gravel were designed as 5, 1, and 6, respectively. The transmitter was 40 mm apart from the receiver, and the finite difference time domain method was used to simulate electromagnetic wave propagation. The used electromagnetic wave was the normalized first derivative of a Gaussian curve with a center frequency of 1.6 GHz. Once the simulation models, i.e., the cavity and gravel cases, are prepared, the transmitter and receiver scan along the soil layer surface with spatial intervals of 20 mm for each model. Figure 7b,c shows the resultant images of the cavity and gravel cases, respectively. As shown in Figure 7b, the $\Delta\theta$ value of the radiated waves expressed by the dashed blue box has the positive value. Similarly, the cavity with the line red box has the positive value, which means in-phase with respect to the $\Delta\theta$ value of the radiated waves. On the other hand, Figure 7c shows the $\Delta\theta$ value of the gravel with the dotted green box is out-of-phase compared with the $\Delta\theta$ value of the radiated waves. Through this precise phase analysis in Phase III, the classification results obtained by considering only shape and amplitude recognition in Phase I can be updated.

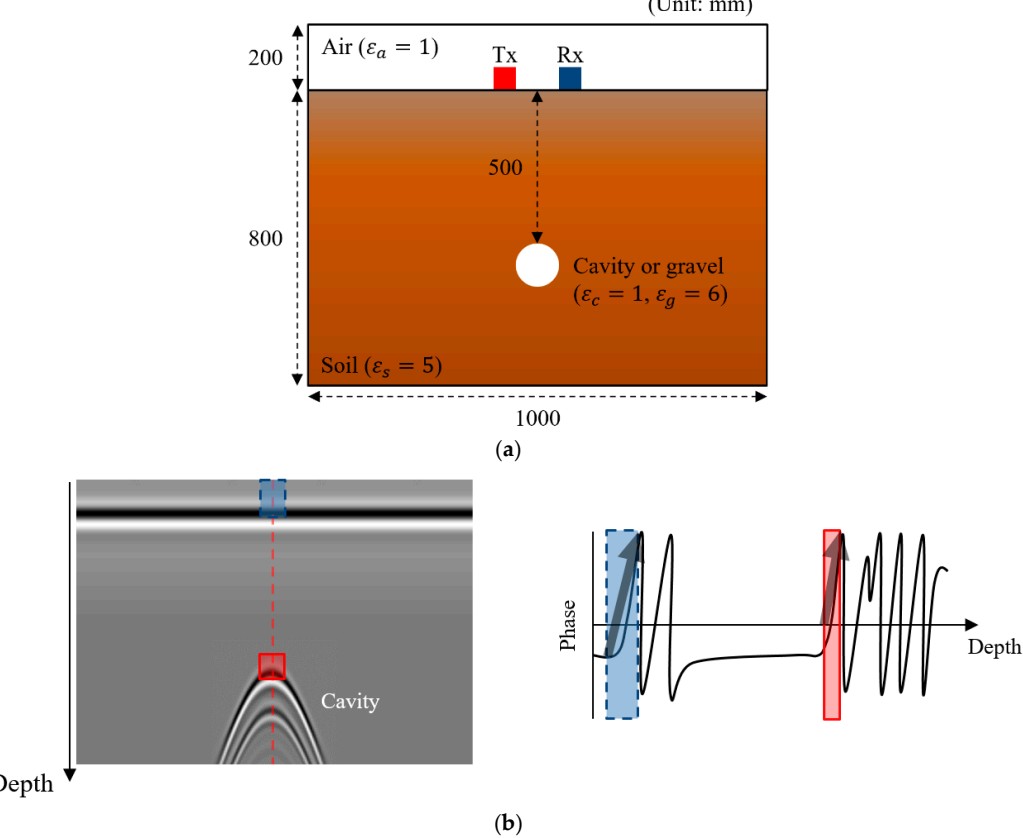

**Figure 7.** *Cont.*

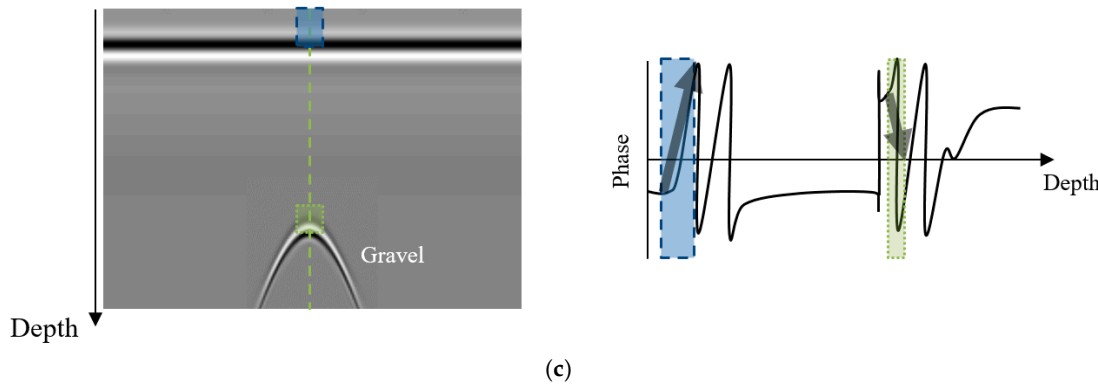

(**c**)

**Figure 7.** Numerical simulation of phase analysis: (**a**) 2D simulation model, (**b**) cavity case, and (**c**) gravel case. The dashed blue box, line red box, and dotted green box in the B-scan images correspond to the locations of the radiated wave, cavity, and gravel boundaries, respectively. (Tx: Transmitter antenna, Rx: Receiver antenna).

## 3. Experimental Validation

The newly proposed UcNet was experimentally validated using 3D GPR data obtained by a multi-channel GPR-mounted van at urban roads in Seoul, South Korea. Figure 8a shows the multi-channel GPR (DXG 1820, 3d-Radar company) [32] has 20 channels transmitting and receiving antennas, which is devised with 0.075 m interval of each channel that is able to cover 1.5 m scanning width at once. The multi-channel GPR designed consists of bow-tie monopole antennas. The multi-channel GPR has a frequency range of 200–3000 MHz with a step-frequency input wave, and the data acquisition system of GeoScopeTM Mk IV (Figure 8b) acquires the GPR data in real time with a time resolution of 0.34 ns and a maximum scanning rate of 13,000 Hz [32].

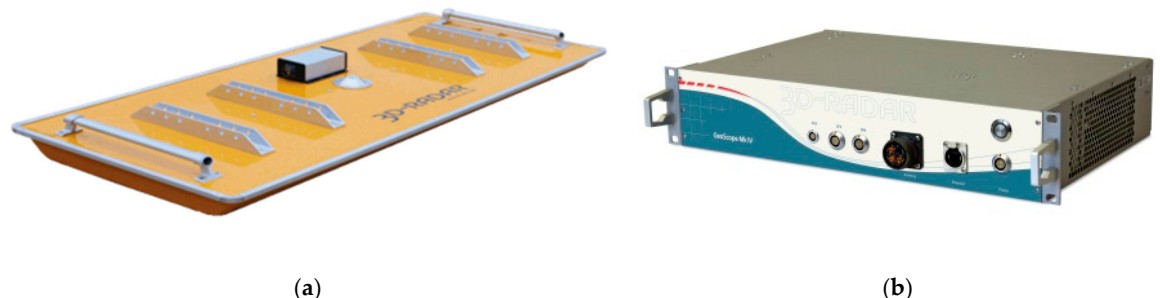

(**a**) (**b**)

**Figure 8.** (**a**) Multi-channel GPR and (**b**) data acquisition system.

As for the training dataset, several tens of kilometers of GPR data, including cavities, manholes, gravels and subsoil background were collected from 17 different regions in Seoul. To clearly confirm the underground objects, the pavement core drilling machine with a portable endoscope was used. Figure 9a shows the multi-channel GPR-mounted van used for field data collection and Figure 9b shows the confirmation process of verifying the underground objects found by the multi-channel GPR-mounted van. The representative confirmed underground cavities and gravels by portable endoscope are shown in Figure 10. A total of 1056 GPR grid images of 256 cavities, 256 manholes, 256 subsoil backgrounds, and 256 gravels cases were used for network training. Here, 20 training epochs and 0.001 initial learning rate were used in this study.

A total of 1056 GPR grid images of 256 cavities, 256 manholes, 256 subsoil backgrounds, and 256 gravels cases, which were not used for UcNet training, were used for blind testing in this study. Figure 11 shows the representative testing 2D GPR grid images of cavities, manholes, subsoil backgrounds, and gravels. As shown in Figure 11b, manholes generally have a distinguishable feature, which has double parabola shape with high intensity compared to the surrounding soil and upper

pavement layer. On the other hand, there is no remarkable feature in B- and C-scan images of the subsoil background case as shown in Figure 11c. However, it can be observed that the B- and C-scan images of the cavity and gravel cases respectively show similar parabola and circular features by comparing between Figure 11a,d.

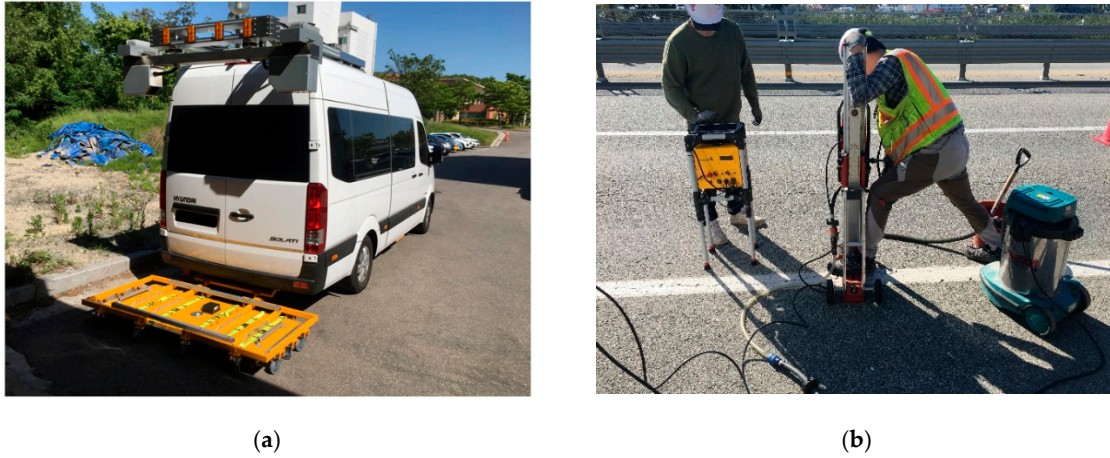

(**a**)                                                           (**b**)

**Figure 9.** In-situ validation tests with (**a**) multi-channel GPR-mounted van and (**b**) core drilling.

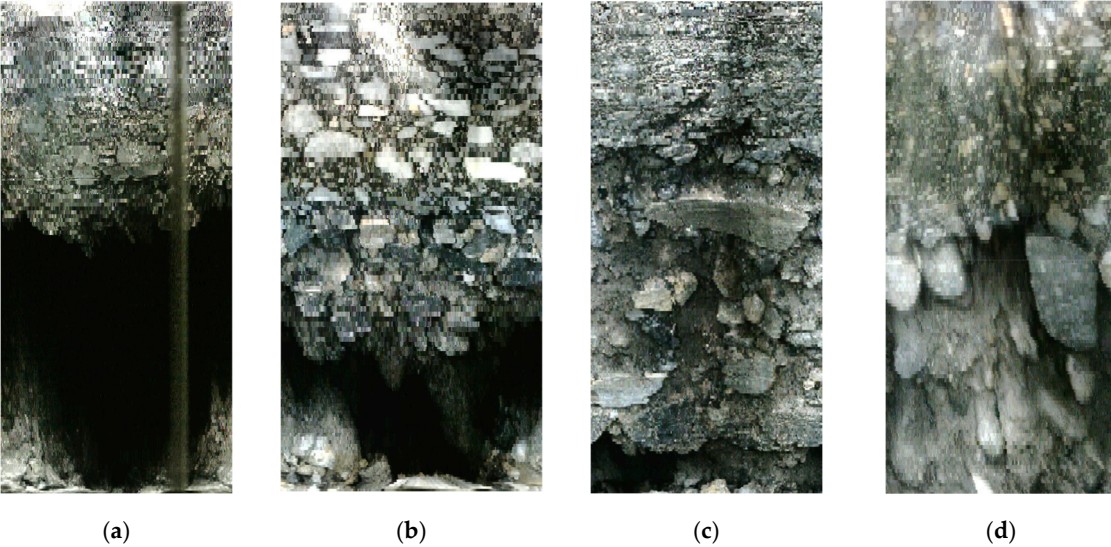

(**a**)                          (**b**)                          (**c**)                          (**d**)

**Figure 10.** Representative core drilling results of (**a**,**b**) cavities and (**c**,**d**) gravels.

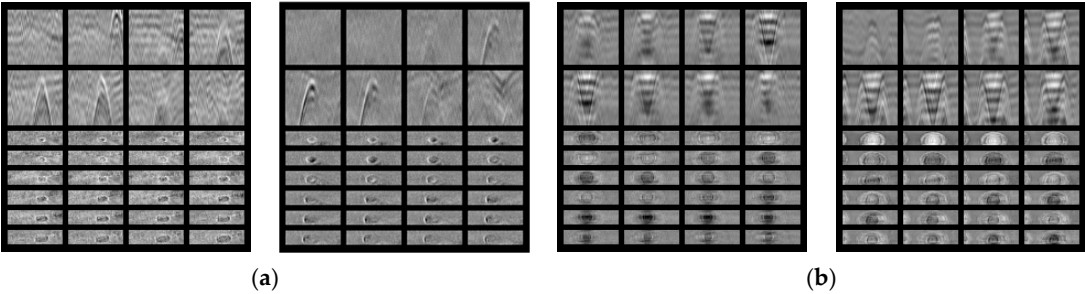

(**a**)                                                           (**b**)

**Figure 11.** *Cont.*

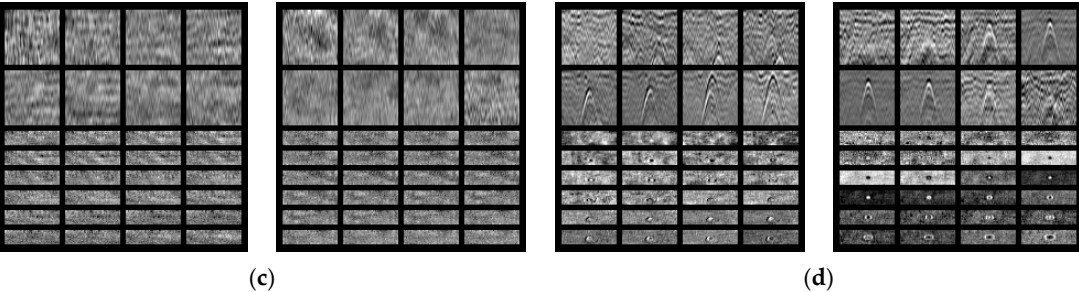

(c)                                                                (d)

**Figure 11.** The representative testing images of (**a**) cavities, (**b**) manholes, (**c**) subsoil backgrounds, and (**d**) gravels.

### 3.1. Conventional CNN-based Underground Object Classification

To validate the effectiveness of UcNet, the two experimental validation results of the conventional CNN and newly developed UcNet were compared. Figure 12 shows the conventional CNN-based underground object classification results. Since Phase I of UcNet is equivalent to the conventional CNN, the processing results up to Phase I were considered as the conventional CNN one. As expected, manhole and subsoil background, which have significant features of 2D GPR grid images, are correctly classified compared with the ground truth confirmed by the portable endoscope. However, 11.74% of cavities and 33.73% of gravels are misclassified as each other due to their similar morphological features. The classification performance of the conventional CNN was evaluated by calculating statistical indices called precision and recall using the following equation:

$$\text{Precision} = \frac{\textit{true positive}}{\textit{true positive} + \textit{false positive}} \quad \text{Recall} = \frac{\textit{true positive}}{\textit{true positive} + \textit{false negative}}. \tag{4}$$

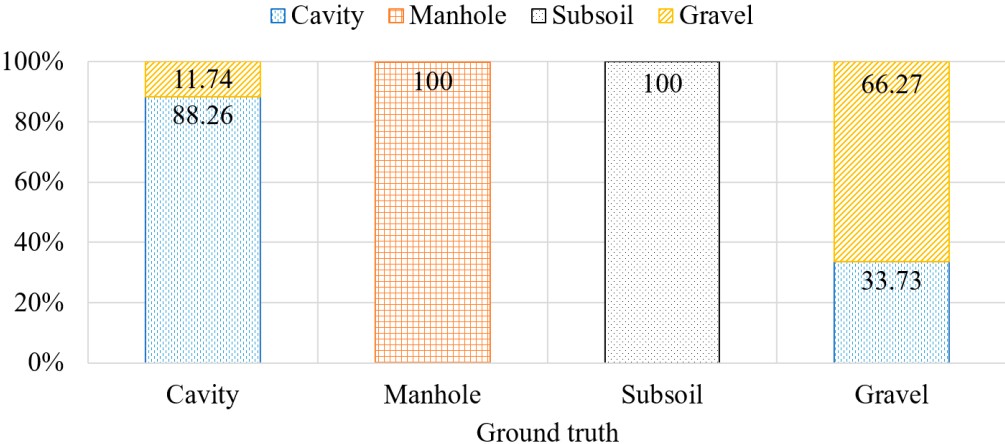

**Figure 12.** The results of conventional convolutional neural network (CNN)-based underground object classification.

Table 1 summarizes the precision and recall values obtained from the conventional CNN results. As for the manhole and subsoil background cases, the precision and recall values are 100%, indicating that they are properly classified by the conventional CNN. On the other hand, 88.26% and 66.27% of the precision values in the cavity and gravel cases, physically meaning that false positive occurs. Similarly, the relatively low recall values of the cavity and gravel cases means the false negative alarm due to the misclassification between the cavity and gravel cases.

**Table 1.** Statistical results obtained from conventional CNN.

| Underground Object | Precision (%) | Recall (%) |
| --- | --- | --- |
| Cavity | 88.26 | 72.36 |
| Manhole | 100 | 100 |
| Subsoil background | 100 | 100 |
| Gravel | 66.27 | 84.95 |

### 3.2. Newly Developed UcNet

From the Phase I results described in Figure 12, Phases II and III were subsequently carried out. Figure 13a–d shows the representative SR B-scan images which are especially misclassified in Phase I. Figure 13a,b indicates the representative cavities cases misclassified as gravels, and Figure 13c,d shows vice versa. The misclassification results show very similar geometric features to each other, but the phase information at the parabola boundaries are distinctive between the cavity and gravel cases. In particular, although the LR GPR B-scan images has ambiguous pixel-level boundary information, the SR images show that much clearer parabola boundary information, making it possible to conduct the precise phase analysis in the subsequent Phase II.

Figure 14a,b indicates the procedure of parabola boundary extraction results with SR and noise removal image corresponding to Figure 13a,b. All parabola boundaries are clearly extracted from the SR images even though unwanted noise and non-parabola features coexist. Similarly, Figure 14c,d shows that the distinctive parabola boundaries are successfully extracted, which correspond to Figure 13c,d.

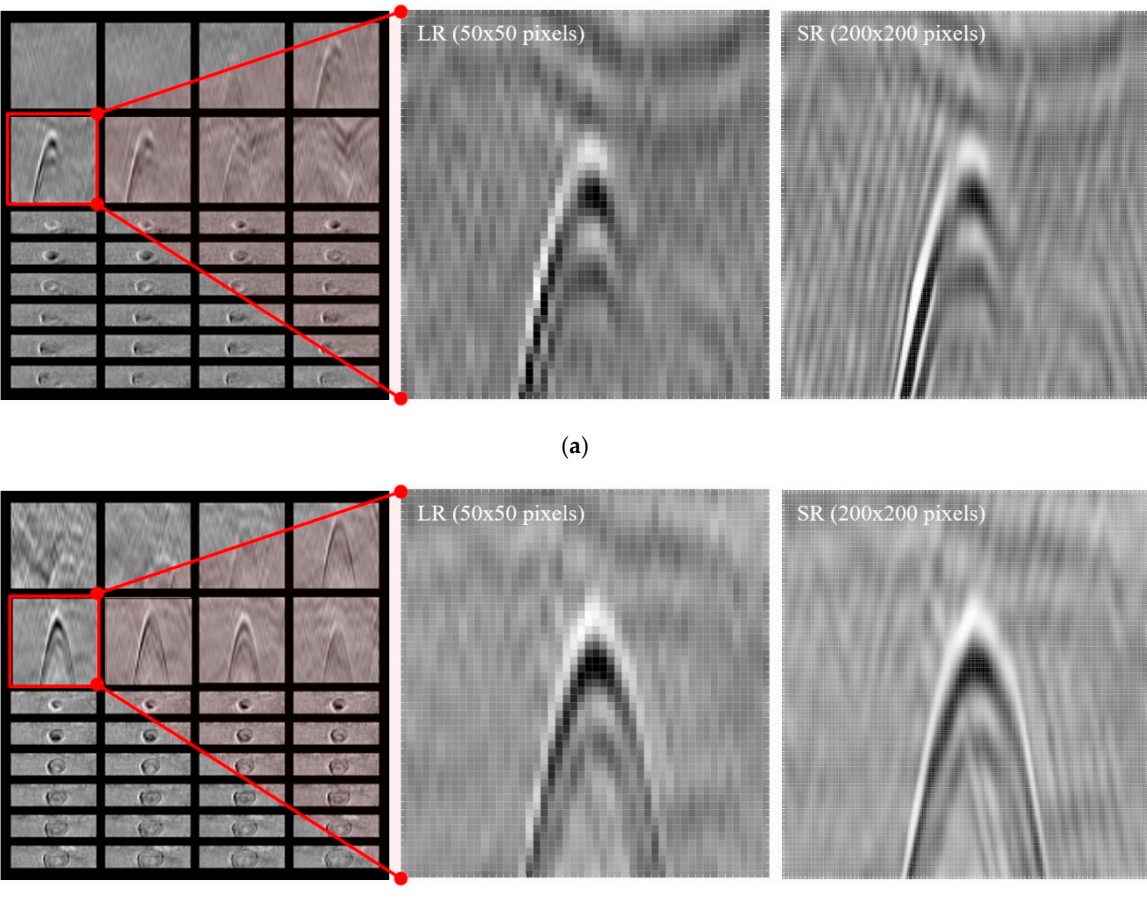

**Figure 13.** *Cont.*

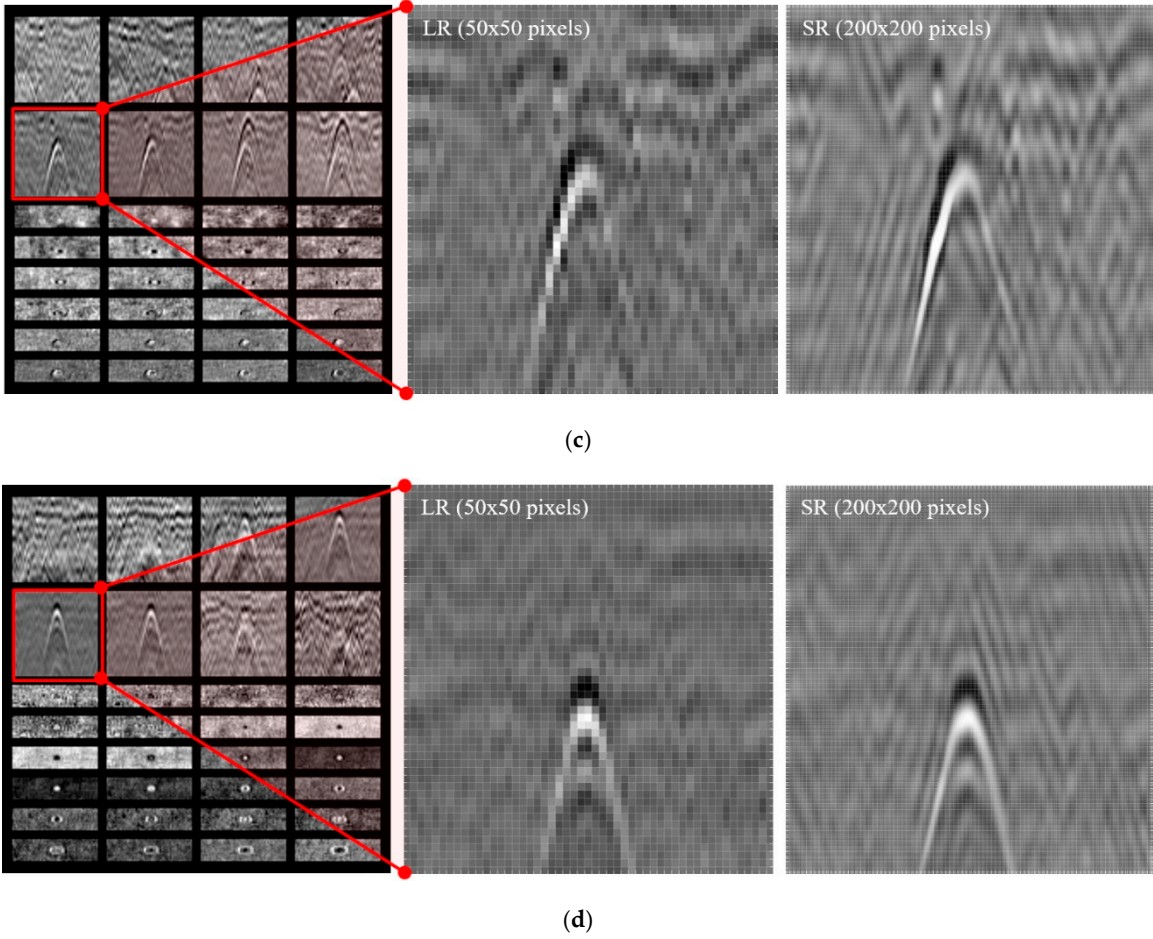

(c)

(d)

**Figure 13.** Representative LR and SR B-scan images of (**a**,**b**) cavity cases misclassified as gravels and (**c**,**d**) gravel cases misclassified as cavities.

Figure 15 shows the phase analysis results corresponding to the extracted parabola boundary information in Figure 14. As shown in Figure 15a, $\Delta\theta$ of the radiated waves has 0.062 which is positive value. Then, the $\Delta\theta$ values of 0.0481 and 0.0336 shown in Figure 15b,c indicate that they can be considered as underground cavities, not gravel. Conversely, the $\Delta\theta$ values of the misclassified cases from gravels to cavities have $-0.0803$ and $-0.1073$ as shown in Figure 15d,e, respectively. These out-of-phase information physically imply the high permittivity of the object in comparison with the surrounding soil, meaning that they are most likely gravel in the designed category of UcNet.

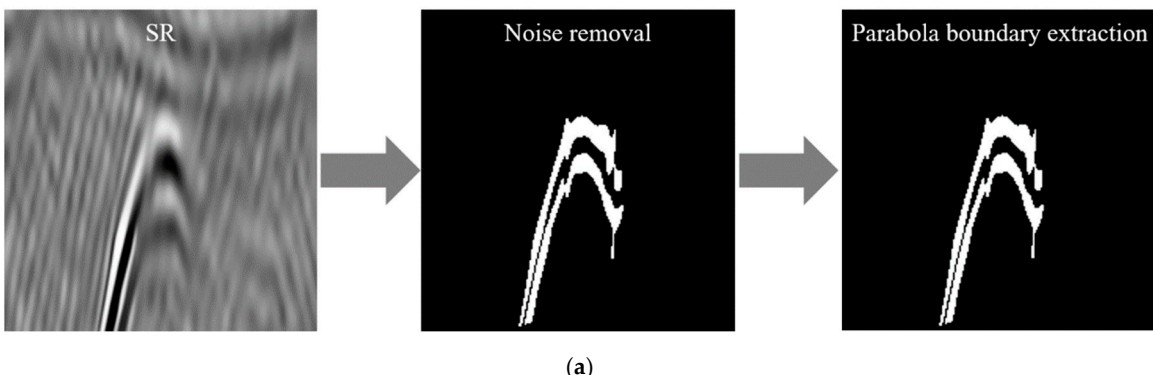

(a)

**Figure 14.** *Cont.*

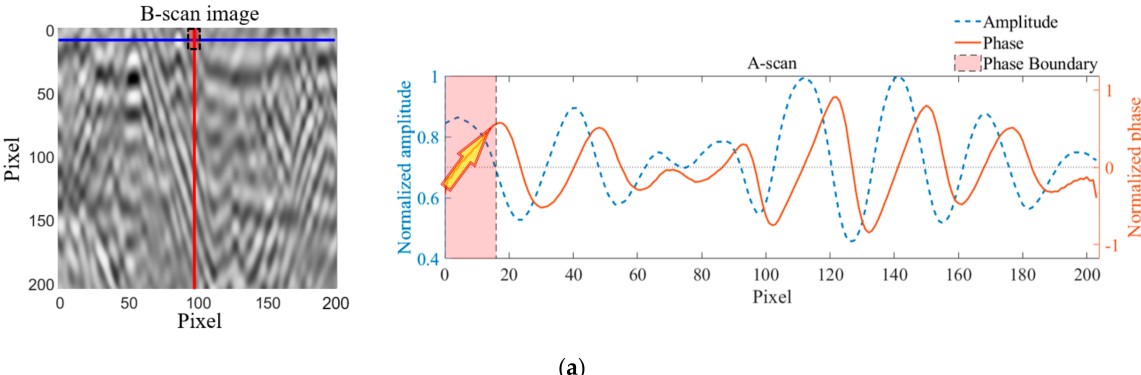

**Figure 14.** The parabola boundary extraction results of (**a**,**b**) cavity cases misclassified as gravels and (**c**,**d**) gravel cases misclassified as cavities.

**Figure 15.** *Cont*.

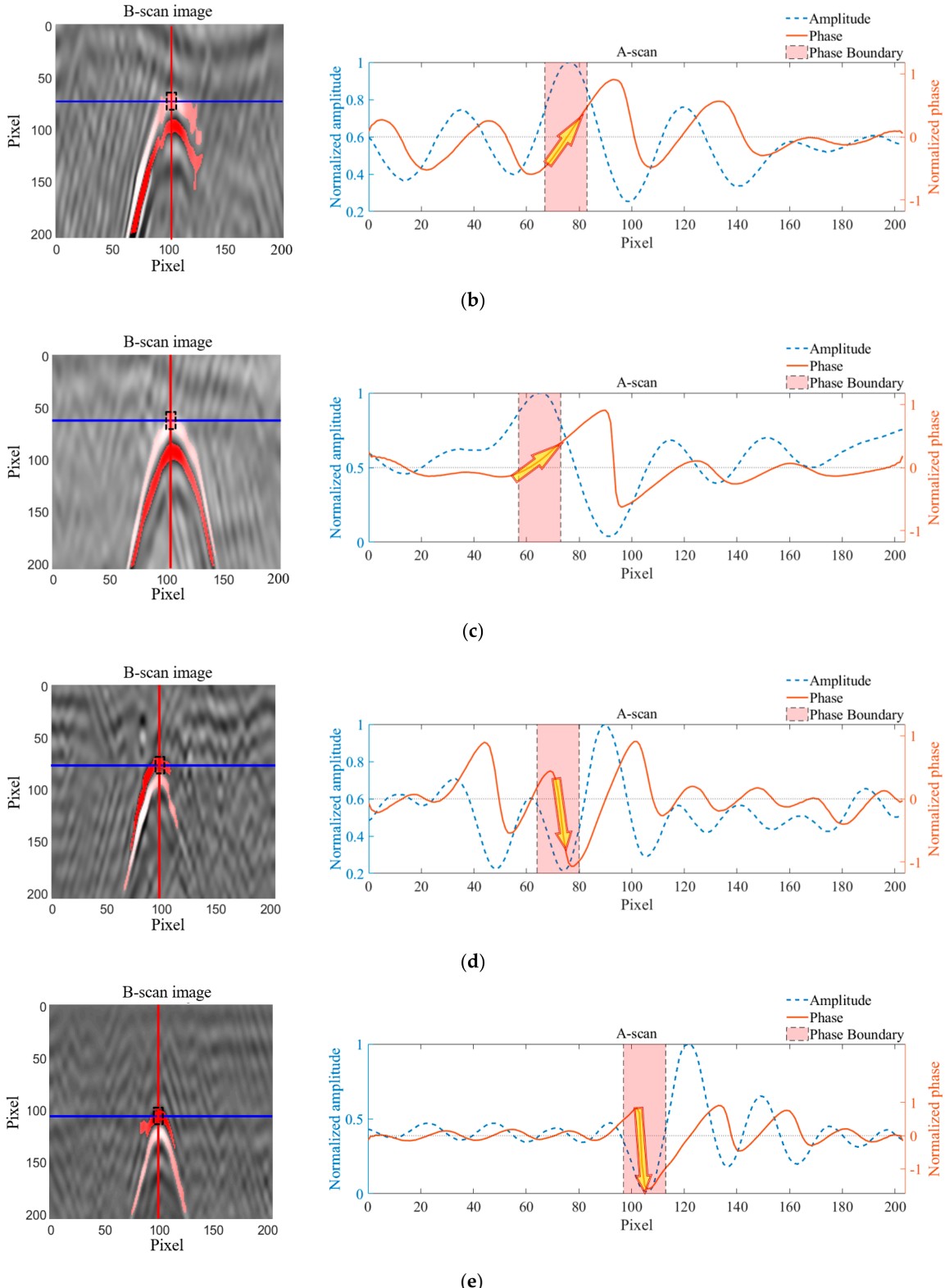

**Figure 15.** Phase analysis results at the extracted boundaries of (**a**) radiated wave, (**b**,**c**) cavities, and (**d**,**e**) gravels, respectively.

Based on the phase analysis results of Figure 15, the object classification results of Figure 11 were updated as shown in Figure 16. It can be easily observed that the misclassified cavities and gravels

are properly updated without false alarms. Since all of misclassification cavity and gravel cases are correctly classified, the statistical precision and recall are increased to 100%.

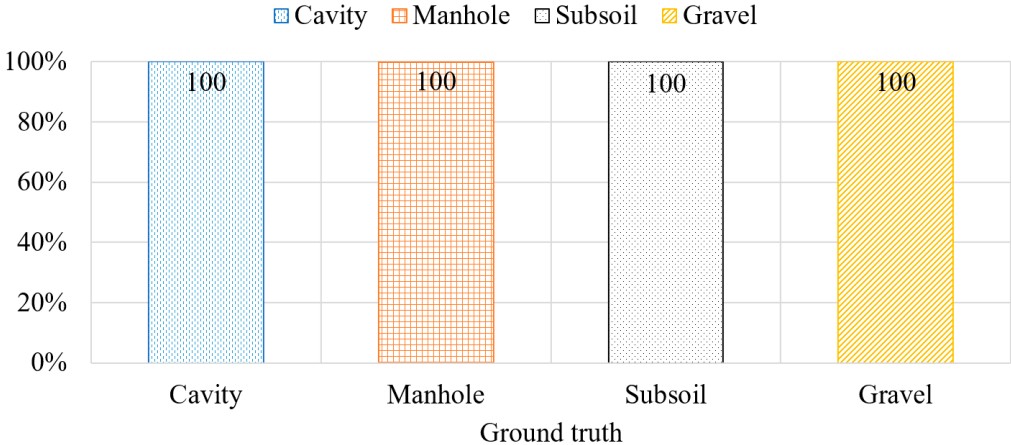

**Figure 16.** Updated underground object classification results using UcNet.

## 4. Conclusions and Discussion

This study newly proposes an underground cavity detection network (UcNet) for enhancing the cavity classification capability. Although convolutional neural networks (CNNs) utilized ground penetrating radar (GPR) triplanar images to classify underground objects, misclassification often occurs due to similar morphological features in B- and C-scan GPR images. This misclassification may lead to a substantial increase of maintenance cost and time. The proposed UcNet overcomes the existing technical hurdle through precise and reliable interpretation of GPR data without expert intervention. In particular, UcNet minimizes the misclassification between cavities and gravel chunks using the conventional CNNs. The effectiveness of the proposed UcNet was experimentally validated using in-situ GPR data obtained on real complex urban areas in Seoul, South Korea. Although the proposed UcNet works well with the validation datasets considered in this study, further investigations on other types of underground objects such as concrete dummies and underground pipes under various in-situ road and underground conditions are warranted. In particular, the authors are now creating our own deep classification network to directly handle 3D GPR data as well as constructing a GPR data library.

**Author Contributions:** M.-S.K. and Y.-K.A. conceived and designed this study. M.-S.K. and N.K. acquired experimental data and performed conventional CNN classification processing. S.B.I., J.-J.L. performed newly proposed UcNet classification processing and analysis. M.-S.K. and Y.-K.A. wrote the entire manuscript and designed the processing code. Y.-K.A. also helped to design the processing code and provided comments.

**Funding:** This work was supported by the National Research Foundation of Korea (NRF) grant funded by the Korea government (MSIT) (2018R1A1A1A05078493).

**Acknowledgments:** The authors would like to thank "Development of Evaluation and Analysis Technologies for Road Sink Research Team" for providing the GPR devices.

**Conflicts of Interest:** The authors declare no conflict of interest.

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
