# Peer review of "3D GPR Image-based UcNet for Enhancing Underground Cavity Detectability"

_remotesensing, doi:10.3390/rs11212545_

Round 1

Reviewer 1 Report

In this paper, a deep learning-based ensemble of networks is used to perform different tasks aiming at classification of underground cavities. The proposal has three main parts: first, it is used an AlexNet to perform a direct classification. Then, a encoder-decoder architecture produces a superresolution version of the input image. Finally, an analysis is performed with traditional vision techniques to produce a final classification. The paper is fairly well written and is easy to follow. Nonetheless, there are some points that must be improved:

- Which is the original resolution of the patches forwarded to the network? The input size of the Phase I is 227x227x3, but the Phase II scales from 50x50 to 200x200. If the original input size is 227x227, then Phase II is meaningless
- Why the input size of Phase I has 3 channels? It seems to me that the images are not colored. What do the 3 channels mean?
- It is unclear how the outputs of Phase I and III are added together in order to produce a final classification. This must be clarified.
- 3.1 shows results of a "Conventional CNN". What does it mean? Which architecture was involved in this experiment? Is it an AlexNet like Phase I?
- If so, I encourage the authors to test a more powerful architecture like InceptionV3 or ResNet50. This would also justify the proposed pipeline.
- The proposal achieved a 100% accuracy, which seems somewhat fishy. The authors should definitely benchmark their approach with some state of the art datasets (they mentioned DIV2K and Urban100) 
- The paper claims that the approach aims to reduce the false positives, but the authors did no efforts to address this particular issue. They increased the accuracy, but they did not perform any action specifically designed to ameliorate the false positives rates.
- The Table I and II seem unnecessary as the information they provide can be easily included in the text.

The authors should carefully address each of these concerns before this manuscript could be considered for publication.

Author Response

Subject: Response to the Reviewer’s Comments for the Paper Submittal to the REMOTE SENSING

Paper Title: GPR Image-based UcNet for Enhancing Underground Cavity Detectability

Responses to the First Reviewer’s Comments

The authors’ responses to the reviewers’ comments are listed below. Each comment is addressed specifically below along with a brief description of how we have modified the manuscript to reflect the respective comments. We appreciate the reviewers’ comments and believe the reviewers’ comments have helped us make this manuscript a better document.

Reviewer #1

Comments to the Authors
In this paper, a deep learning-based ensemble of networks is used to perform different tasks aiming at classification of underground cavities. The proposal has three main parts: first, it is used an AlexNet to perform a direct classification. Then, a encoder-decoder architecture produces a super resolution version of the input image. Finally, an analysis is performed with traditional vision techniques to produce a final classification. The paper is fairly well written and is easy to follow. Nonetheless, there are some points that must be improved:

Please consider the following aspects to improve the paper:

- Which is the original resolution of the patches forwarded to the network? The input size of the Phase I is 227x227x3, but the Phase II scales from 50x50 to 200x200. If the original input size is 227x227, then Phase II is meaningless.

Response: The authors appreciate the reviewer’s comments. The following figure and explanations have been added in the revised manuscript (line 149-153) for readers’ better understanding.

‘The reconstructed GPR grid image is comprised of 8 B-scan and 24 C-scan images. The original sizes of B- and C-scan images are 50 × 50 and 50 × 13, respectively. To proceed Phase II, the 5th representative B-scan image is selected from each reconstructed GPR 2D grid image. Since the resolution of the selected B-scan image is not enough for Phase II as shown in Figure 6, the subsequent SR image generation process is necessary.’

Please see the attachment. Figure 6 is added in the attachment. 

- Why the input size of Phase I has 3 channels? It seems to me that the images are not colored. What do the 3 channels mean?

Response: The 3 channels means RGB, i.e. red, green and blue color components. Generally, the image classification networks such as AlexNet have utilized 3 channels RGB information, because more distinctive image features can be secured. Since the proposed UcNet was developed by transfer learning using AlexNet due to the limited in-situ GPR data for proper hyper parameter training, we intentionally made GPR data to RGB format. As the follow-up study, we now collecting more various GPR data for developing our own classification network, which hopefully enhance the classification performance.

- It is unclear how the outputs of Phase I and III are added together in order to produce a final classification. This must be clarified.

Response: Thanks for the reviewer’s point out. The purpose of the proposed UcNet is focused on the underground cavity detectability enhancement, not all data classification, as the first trial study. The only cavity-like features in B-scan images, which have parabola morphographical features, are only proceeded to Phases II and III. Thus, the final output of Phase III is used for the final cavity detection results, although Figure 16 of the revised manuscript shows the classification results to fairly compare the Figure 12 results. however, as the reviewer pointed out, more robust network which can classify arbitrary underground objects needs to be developed as the follow-up study.

- 3.1 shows results of a "Conventional CNN". What does it mean? Which architecture was involved in this experiment? Is it an AlexNet like Phase I?

Response: As the reviewer pointed out, the conventional CNN means Phase I, which was constructed by transfer learning of AlexNet in this study. To compare the performance of the proposed UcNet with the AlexNet one, which was previously used for underground cavity detection [27], the terminology of ‘conventional CNN’ was used in this manuscript.

- If so, I encourage the authors to test a more powerful architecture like InceptionV3 or ResNet50. This would also justify the proposed pipeline.

Response: The authors appreciate the valuable comments. As the reviewer aware, it has been well known that the transfer learning shows very effective image classification performance like Inception V3 or ResNet50. The deeper network such as Inception V3 and ResNet50, which have 48 and 50 layers, needs a large number of datasets to tune all the hyper parameters which influence the classification performance [Ref #1]. Thus, in this study, due to limited number of GPR data, a transfer learning approach has been employed with the 8 layers pre-built AlexNet which is a relatively simpler deep CNN. As the follow-up study, the authors are now developing an own deep classification network to handle three-dimensional GPR data as well as constructing GPR data library.

[Ref #1] Kafedziski V. et al. Detection and classification of land mines from ground penetrating radar data using faster R-CNN, 26th Telecommunications forum TELFOR 2018.

- The proposal achieved a 100% accuracy, which seems somewhat fishy. The authors should definitely benchmark their approach with some state of the art datasets (they mentioned DIV2K and Urban100).

Response: The authors appreciate the reviewer’s point out. The authors tried to focus on the fact that the misclassification of the conventional CNN is significantly reduced by using the proposed UcNet. To experimentally validate it, we tried to intentionally find and collect the gravel cases, which was especially similar to the cavity ones. Note that these cases are the worst cases in terms of data classification. As the reviewer aware, the corresponding in-situ dataset collection was quite difficult in complex urban areas, which was rare opportunity. Due to the limited test datasets, it is difficult to be generalized, but the 100 % accuracy results were obtained by the proposed UcNet within our test datasets. As the follow-up study, further investigations on other types of underground objects such as concrete dummy and underground pipe under various in-situ road and underground conditions are warranted.

As for the benchmark datasets of DIV2K and Urban 100 were only used to train and validate for the SR network. These datasets are not involved in the final classification decision of UcNet.

- The paper claims that the approach aims to reduce the false positives, but the authors did no efforts to address this particular issue. They increased the accuracy, but they did not perform any action specifically designed to ameliorate the false positives rates.

Response: The authors thank to the reviewer’s comments. As the reviewer suggested, more comprehensive interpretations of each step result with additional figures have been added in the revised manuscript. In particular, the statistical indices such as precision and recall have been added in Table 1 of the revised manuscript.

- The Table I and II seem unnecessary as the information they provide can be easily included in the text.

Response: It has been revised accordingly.

‘Total 1,056 GPR grid images of 256 cavity, 256 manhole, 256 subsoil background and 256 gravel cases, which were not used for UcNet training, are used for blind testing in this study.’

Reviewer 2 Report

Title: 3D GPR Image-based UcNet for Enhancing Underground Cavity Detectability

Overall comment:

The paper presents a new CNN network (UcNet) for enhancing underground cavity detectability. The proposed UcNet is developed based on phase analysis of super-resolution GPR images to minimize misclassification of underground cavity. Experiments on real roads are conducted and show a promising performance of the UcNet based method. In overall, the paper is well organized with a good structure as well as it is readable. There are a few comments for the consideration of revision.

What would be primary differences and academic contributions compared to the authors’ prior studies such as the references [26-27]? Does the integration with the phase analysis introduced in [27] provide a meaningful value to be claimed as a primary academic contribution? Please clarify. In terms of description of data processing steps, the authors are suggested to provide the detailed explanation of the sub-steps for each Phase for readers although the updated CNN steps are well recognized. This comment is applied to Section 3 Experimental Validation. More comprehensive interpretation of the results are necessary and more discussion points on the results would enhance the quality of this paper. It is suggested to include impacts/benefits of the proposed method in Conclusion section.

Minors:

Figure 9 image resolution is low, which should be significantly increased.

Author Response

Subject: Response to the Reviewer’s Comments for the Paper Submittal to the REMOTE SENSING

Paper Title: GPR Image-based UcNet for Enhancing Underground Cavity Detectability

Responses to the Second Reviewer’s Comments

The authors’ responses to the reviewers’ comments are listed below. Each comment is addressed specifically below along with a brief description of how we have modified the manuscript to reflect the respective comments. We appreciate the reviewers’ comments and believe the reviewers’ comments have helped us make this manuscript a better document.

Reviewer #2

Comments to the Authors
The paper presents a new CNN network (UcNet) for enhancing underground cavity detectability. The proposed UcNet is developed based on phase analysis of super-resolution GPR images to minimize misclassification of underground cavity. Experiments on real roads are conducted and show a promising performance of the UcNet based method. In overall, the paper is well organized with a good structure as well as it is readable. There are a few comments for the consideration of revision.

- What would be primary differences and academic contributions compared to the authors’ prior studies such as the references [26-27]? Does the integration with the phase analysis introduced in [27] provide a meaningful value to be claimed as a primary academic contribution? Please clarify.

Response: The authors appreciate to the reviewer’s comments. The following explanations were clearly mentioned in the introduction section (line 65-82) of the manuscript. The authors strongly believe that the proposed technique combining the CNN-based super-resolution GPR image classification and phase analysis would be potentially meaningful for academic contribution and real-world applications.

Although the developed CNN using the combination of B- and C-scan images increased the classification accuracy compared to the conventional CNN using only B-scan image [26, 27], it was turned out that it still has difficulty to differentiate underground cavity from chunk of gravel especially in complex urban areas due to their similar GPR reflection features. To address the misclassification issue caused by the underground chunk of gravel, Park et al. recently proposed a phase analysis technique of GPR data [16]. However, the temporal and spatial resolutions of the 3D GPR data are often insufficient for the precise phase analysis. Since the phase analysis results highly depends on a few pixel differences of the GPR images, the lack of GPR image resolution may cause false alarms during the phase analysis.

In this paper, 3D GPR image-based underground cavity detection network (UcNet), which consists of CNN and the phase analysis of super-resolution (SR) GPR images, is newly proposed to enhance underground cavity detectability. By retaining the advantages of both CNN and phase analysis, underground cavity can be automatically classified with minimized false alarms. In particular, a deep learning-based SR network used for GPR image resolution enhancement significantly reduce the misclassification between the underground cavity and chunk of gravel. To examine the performance of the proposed UcNet, comparative study results on cavity detectability between the conventional CNN and the newly proposed UcNet are presented using in-situ 3D GPR data obtained from complex urban roads in Seoul, South Korea.

- In terms of description of data processing steps, the authors are suggested to provide the detailed explanation of the sub-steps for each Phase for readers although the updated CNN steps are well recognized. This comment is applied to Section 3 Experimental Validation. More comprehensive interpretation of the results are necessary and more discussion points on the results would enhance the quality of this paper. It is suggested to include impacts/benefits of the proposed method in Conclusion section.

Response: Thanks for the reviewers’ valuable comments. As the reviewer suggested, more comprehensive interpretations of each step result with additional figures have been added in the revised manuscript. Also, the benefits of the proposed technique have been more clearly added in the conclusion section of the revised manuscript accordingly.

- Figure 9 image resolution is low, which should be significantly increased.

Response: It has been revised accordingly.

Reviewer 3 Report

Thank you for the opportunity to review the article titled 3D GPR Image-based UcNet for Enhancing Underground Cavity Detectability.

The paper is well written and well structured. The article is scientifically important and with its novelty contributes to the remote sensing literature. The paper represents the use of a convolutional neural network for enhancing ground-penetrating radar images.  

The introduction is well structured and the importance/problem of the study is well presented. The introduction also gives a brief literature review, and afterward, the aim of the study is presented. The study has been conducted in South Korea. 

In my opinion, line 83-85 needs to be restructured. 

The methodological part presented in section 2. Development of UcNet, is clear, well written and well structured. 

In the Experimental validation, the authors compared the developed UcNet with conventional methods, and the results showed that the new developed method is more effective and accurate.  

There is a clear leak of discussion. The author should discuss the findings in the study and compare it with similar studies or emphasize the importance of the results. Section 4 is a conclusion. 

Author Response

Subject: Response to the Reviewer’s Comments for the Paper Submittal to the REMOTE SENSING

Paper Title: GPR Image-based UcNet for Enhancing Underground Cavity Detectability

Responses to the Third Reviewer’s Comments

The authors’ responses to the reviewers’ comments are listed below. Each comment is addressed specifically below along with a brief description of how we have modified the manuscript to reflect the respective comments. We appreciate the reviewers’ comments and believe the reviewers’ comments have helped us make this manuscript a better document.

Reviewer #3

Comments to the Authors
Thank you for the opportunity to review the article titled 3D GPR Image-based UcNet for Enhancing Underground Cavity Detectability. The paper is well written and well structured. The article is scientifically important and with its novelty contributes to the remote sensing literature. The paper represents the use of a convolutional neural network for enhancing ground-penetrating radar images. The introduction is well structured and the importance/problem of the study is well presented. The introduction also gives a brief literature review, and afterward, the aim of the study is presented. The study has been conducted in South Korea. In my opinion, line 83-85 needs to be restructured. The methodological part presented in section 2. Development of UcNet, is clear, well written and well structured. In the Experimental validation, the authors compared the developed UcNet with conventional methods, and the results showed that the new developed method is more effective and accurate. There is a clear leak of discussion. The author should discuss the findings in the study and compare it with similar studies or emphasize the importance of the results. Section 4 is a conclusion.

Response: The authors appreciate the reviewer’s valuable comments. As the reviewer suggested, the line 83-85 is reconstructed in the revised manuscript as follows:

“This paper is organized as follows. Section 2 explains the proposed UcNet, which is comprised of CNN, SR image generation and phase analysis. Then, 3D GPR data collection procedure from urban roads and experimental validation are described in section 3. In particular, the comparative study results between the conventional CNN and newly proposed UcNet are addressed. Finally, section 4 concludes the paper with brief discussion.”

As the reviewer suggested, much more comprehensive interpretations of each step result with additional figures have been added in the revised manuscript.

Round 2

Reviewer 1 Report

The authors successfully addressed all the issues I raised in my previous review.
Now I feel the paper is ready for publication.